# Impact of an Exotic Invasive Pest, *Spodoptera frugiperda* (Lepidoptera: Noctuidae), on Resident Communities of Pest and Natural Enemies in Maize Fields in Kenya

Bonoukpoè Mawuko Sokame [1,2], Boaz Musyoka [1], Julius Obonyo [1], François Rebaudo [3], Elfatih M. Abdel-Rahman [1], Sevgan Subramanian [1], Dora Chao Kilalo [2], Gérald Juma [4] and Paul-André Calatayud [1,3,*]

1 International Centre of Insect Physiology and Ecology (ICIPE), Nairobi P.O. Box 30772-00100, Kenya; bsokame@icipe.org (B.M.S.); bmusyoka@icipe.org (B.M.); jobonyo@icipe.org (J.O.); eabdel-Rahman@icipe.org (E.M.A.-R.); ssubramania@icipe.org (S.S.)
2 Department of Plant Science and Crop Protection, University of Nairobi, Kangemi, Nairobi P.O. Box 29053-00625, Kenya; ngachalor@gmail.com
3 IRD, Université Paris-Saclay, CNRS, UMR Évolution, Génomes, Comportement et Écologie, 91198 Gif-sur-Yvette, France; francois.rebaudo@ird.fr
4 Department of Biochemistry, University of Nairobi, Nairobi P.O. Box 30197-00100, Kenya; gjuma@uonbi.ac.ke
* Correspondence: pcalatayud@icipe.org or paul-andre.calatayud@egce.cnrs-gif.fr

**Abstract:** The interactions among insect communities influence the composition of pest complexes that attack crops and, in parallel, their natural enemies, which regulate their abundance. The lepidopteran stemborers have been the major maize pests in Kenya. Their population has been regulated by natural enemies, mostly parasitoids, some of which have been used for biological control. It is not known how a new exotic invasive species, such as the fall armyworm (FAW), *Spodoptera frugiperda* (Lepidoptera, Noctuidae), may affect the abundance and parasitism of the resident stemborers. For this reason, pest and parasitism surveys have been conducted, before and after the FAW invaded Kenya, in maize fields in 40 localities across 6 agroecological zones (AEZs) during the maize-growing season, as well as at 3 different plant growth stages (pre-tasseling, reproductive, and senescence stages) in 2 elevations at mid-altitude, where all maize stemborer species used to occur together. Results indicated that the introduction of the FAW significantly correlated with the reduction of the abundance of the resident communities of maize stemborers and parasitoids in maize fields; moreover, the decrease of stemborer density after the arrival of FAW occurred mostly at both reproductive and senescent maize stages. It also suggests a possible displacement of stemborers by FAW elsewhere; for example, to other cereals. However, since this study was conducted only three years after the introduction of the FAW, further studies will need to be conducted to confirm such displacements.

**Keywords:** fall armyworm; lepidoptera stemborers; *Busseola fusca*; *Sesamia calamistis*; *Chilo partellus*; parasitoids; biological control

## 1. Introduction

The native stemborers, *Busseola fusca* (Fuller) and *Sesamia calamistis* Hampson (Lepidoptera: Noctuidae), and the invasive stemborer, *Chilo partellus* (Swinhoe) (Lepidoptera: Crambidae), are pests of maize and sorghum in East Africa [1]. The three stemborers occur as single or mixed species communities [2–5], with community structure varying with locality, altitude, and season. *Busseola. fusca* is generally the dominant species in the highlands, *C. partellus* dominates in the lowlands [6,7], and *S. calamistis* occurs at all altitudes [8]. These stemborer species often occur as a mixed community of all three species in mid-altitude regions [4,9,10]. These pests share the same resource (i.e., maize stems), and competition is high [11]. Both intra- and interspecific competition has been observed between *B. fusca*,

*S. calamistis*, and *C. partellus*, with stronger interspecific interaction recorded between the noctuids and the crambid than between the two noctuids [12].

Several studies have documented parasitoids associated with the three stemborers in the different AEZs [13–16]. In cultivated habitats in Kenya, the most common parasitoids of all three species are the larval parasitoids *Cotesia flavipes* Cameron and *Cotesia sesamiae* (Cameron) (Hymenoptera: Braconidae), followed by the pupal parasitoids *Xanthopimpla stemmator* (Hymenoptera: Ichneumonidae) and *Pediobius furvus* Gahan (Hymenoptera: Eulophidae) and the tachinid *Siphona* sp. [13–16]. Among these, the larval parasitoid *C. flavipes*, which was introduced from Asia for classical biological control of *Chilo partellus* [17,18], and *C. sesamiae* are the most common parasitoids of stemborers infesting maize in East and Southern Africa [1]. They have been collected from all three stemborer species in both cultivated and wild habitats [13–16,19]. The overall parasitism rate of stemborers ranges from 0 to 58% in western Kenya and from 0 to 26% in the Coastal region [20].

This community of stemborers and parasitoids might be disturbed by the recent introduction of the fall armyworm (FAW), *Spodoptera frugiperda* J.E. Smith (Lepidoptera: Noctuidae), from the Americas into sub-Saharan Africa, where it has invaded most countries and caused severe damage in maize fields [21,22]. Recent estimates in Zimbabwe [23], Ethiopia [24], and Kenya [25] indicated between 11.5 and 30% yield losses due to the FAW. In Kenya, this pest was first reported in the western region in 2017, and by the early cropping season in 2018, was confirmed throughout the country [26]. Stemborer larvae mainly feed on young leaves until the third instar, and thereafter, mainly on maize stems. FAW larvae, on the other hand, feed on leaves during the maize plant vegetative stage, and especially on the central leaves in the whorl [27,28]. In addition, in maize fields at tasseling stage, FAW larvae can be found feeding on the tassels and subsequently on the ear, silk, cob, and even in stemborer's holes [27,29,30]. FAW and stemborer larvae may, therefore, interact by sharing the same niche at young developmental stages and even when the stemborer larvae migrate from leaves to stems.

The present study aims at evaluating how the presence of FAW affects the abundance of the stemborer species and their parasitism rates in Kenya using maize fields surveyed before and after the introduction of the FAW. The specific objectives of this study were to evaluate (i) the abundance of FAW in different maize AEZs of Kenya; (ii) how the introduction of FAW and its abundance affected stemborer density and parasitism in different AEZs; (iii) how the introduction of FAW and its abundance determined stemborer density in different plant phenological stages.

## 2. Materials and Methods

*2.1. Effect of FAW Introduction and Its Abundance on Stemborer Density and Parasitism across the Maize Agroecological Zones (AEZs) of Kenya*

Sampling was carried out before and after the introduction of the FAW in maize fields in 40 localities situated in the 6 maize AEZs described by [31] in Kenya (Figure 1). The localities ranged from the lowlands in the coastal region (sea level) to the highlands in the western region (2343 m asl) of Kenya. Sampling in the same selected locations was done between 2012 and 2016, before the introduction of the FAW, and between 2018 and 2019, after the introduction of the FAW.

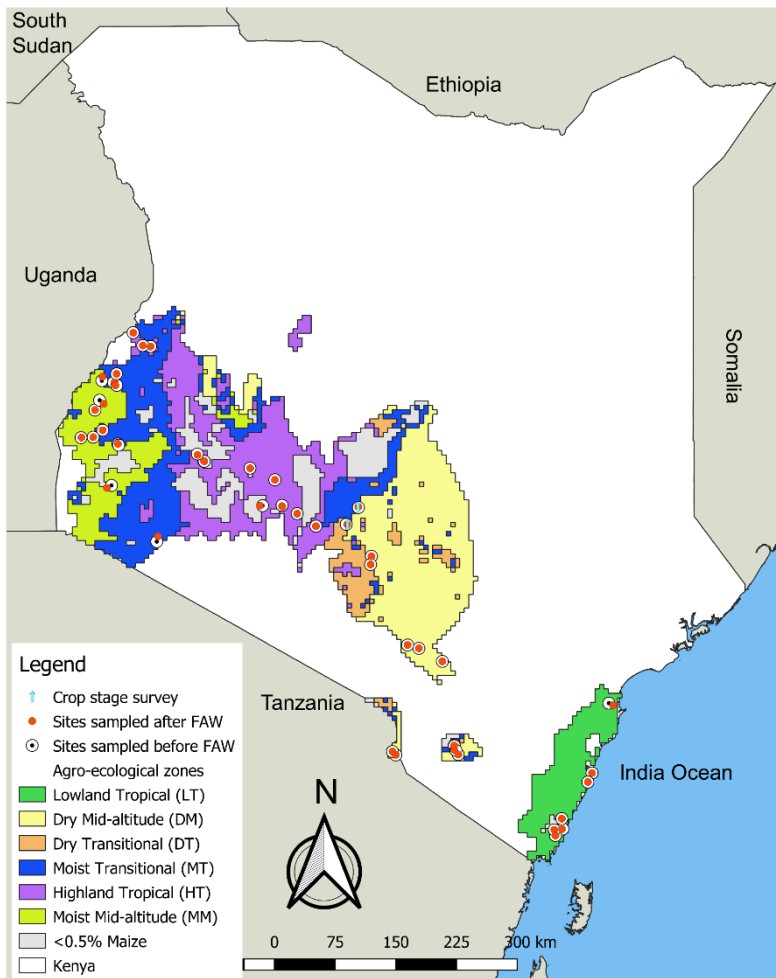

**Figure 1.** Map of sampling localities before and after the introduction of the FAW in the different agro-ecological zones of Kenya.

Samplings followed the protocol described by [32]. Between 5 to 8 locations with 10 fields surveyed per location were surveyed in each AEZs. A total of 100 maize plants were systematically inspected in each field following a zig-zag pattern. All infested plants were collected from the field and dissected to record stemborer and FAW larvae. Species richness and abundance as well as parasitism for species were computed.

All collected stemborer and FAW larvae were reared on an artificial diet developed by [33] and by [34], respectively, in cylindrical glass vials (8.5 cm × 2.5 cm) plugged with cotton wool and kept under ambient conditions in the laboratory (25 ± 1 °C; 67 ± 4% relative humidity) until pupation or until parasitoid emerged. The pupae were kept in separate plastic containers (16 cm × 10 cm) closed with perforated plastic lids until adult emergence for species identification. In case of parasitism, the emerged parasitoids were conserved in 70% ethanol in glass vials (2.5 cm in diameter and 7.5 cm high) for species morphological identification [35–37] in collaboration with the Biosystematics Unit of International Centre of Insect Physiology and Ecology (*icipe*).

Stemborer abundance and FAW abundance were expressed as the total number of larvae and pupae of all stemborer species or of the FAW recorded per 100 maize plants sampled. Parasitism was assessed as the proportion of parasitized larvae and pupae among the total number of larvae and pupae of all stemborer species recorded. The generalized linear mixed model (GLMM) with a Poisson distribution was performed using the lme4 R package [38] on the distribution and abundance of the FAW after its invasion, with AEZs as a fixed effect and Fields/Locations as random effects to take into account



the pseudoreplication. To analyze whether the FAW affects the abundance/density of stemborers and their parasitism, the stemborer species were considered together, along with their parasitism. GLMM with a Poisson distribution was also performed using the lme4 R package [38] and two levels of analysis were considered: (i) the impact of FAW arrival on stemborer abundance and their parasitism, where the FAW (before/after) and AEZs were the fixed effects and Fields/Locations were random effects of field nested within locality; (ii) how the FAW abundance affects stemborers and their parasitism, where FAW abundance (after FAW invasion only) and AEZs were the fixed effects and Fields/Locations were random effects. Where significant difference was obtained, pairwise comparison was made using the least squares means and adjusted Tukey multiple comparison procedure ($\alpha$ = 0.05) in lsmeans and multicompview packages, respectively [39,40]. The proportions of single and multi-species infestations of Figure 4 before and after the introduction of FAW periods were compared using a proportion Z-test.

*2.2. Effect of FAW Introduction and Its Abundance on Stemborer Density across Different Maize Phenological Stages*

For this study, a season-long monitoring survey was conducted before and after the introduction of the FAW in maize fields of two localities at mid-altitude (Figure 1), namely Makutano and Murang'a, where B. fusca, S. calamistis, and C. partellus were known to occur together on maize [4,8–10,41,42]. The surveys were carried out during the cropping season in Makutano (0°43.616′ S, 37°16.373′ E, 1150 m asl), where C. partellus and S. calamistis co-infest maize, and in Murang'a (0°55.387′ S, 37°09.004′ E, 1500 m asl), where B. fusca and S. calamistis are present [19,42]. Mean annual rainfall is 981 mm and 1195 mm and mean annual temperatures are 21.2 °C and 20 °C in Makutano and Murang'a, respectively [19,42]. Mean annual relative humidity ranges from 50 to 72% at both sites. Both sites are characterized by a bimodal rainfall distribution with two cropping seasons, April to June and October to December, with a dry season in between [19,42].

A popular maize variety called Duma 43 (Simlaw, Kenya Seed Company, Nairobi, Kenya) that takes three months to mature was provided to selected farmers at each locality to minimize the effect of plant variety on insect pest infestation. Sampling was done in 2017, before the introduction of the FAW, and in 2018, after the introduction of the FAW, in three fields of each locality.

During the long and short rainy seasons of each year, every field in each locality was sampled twice at two-week intervals and at three different plant growth stages (pre-tasseling, reproductive, and senescence stages) to identify the phenological stages where interactions between maize stemborers and the FAW are likely to occur. Six surveys were undertaken in each maize field during each cropping season. At the pre-tasseling stage, the "W" scouting pattern was used for sampling [34], whereas at reproductive and senescence stages, when plants were taller and the "W" scouting pattern became difficult to use, the "Ladder" scouting pattern was used [34]. A total of 100 maize plants were systematically inspected in each field during each sampling period. Maize plants with damage symptoms were uprooted and dissected to recover larvae and pupae from the stems, whorls, tassels, and ears. The collected larvae and pupae were counted according to species and then placed individually in glass vials (8.5 × 2.5 cm), given an artificial diet, and brought into the laboratory for rearing until adult stage to confirm species identification or recovery of parasitoids in case of parasitism. Species richness, abundance, as well as parasitism for species were computed.

Data were analyzed using GLMM with a Poisson distribution with the lme4 R package [38], where FAW (before/after) or FAW abundance (after FAW invasion only) and plant stages were fixed effects and Fields/Locations were random effects. Where significant differences were obtained, a pairwise comparison was made using the least squares means and adjusted Tukey multiple comparison procedure ($\alpha$ = 0.05) in lsmeans and multicompview packages, respectively [39,40]. All analyses were performed with R software, version 4.0 [43].

## 3. Results

### 3.1. Distribution and Abundance of the FAW in the Different AEZs of Kenya

The fall armyworm occurred in all sampled areas in the different AEZs but in varying abundance (Figure 2; Tables A1 and A2, Appendix A). Larval abundance of the FAW was significantly higher in lowland tropical areas, while lower abundance was recorded in highland tropical areas as presented in Figure 2 (GLMM result: $z$ value = $-5.197$, $p < 0.0001$; Supplementary Table S1).

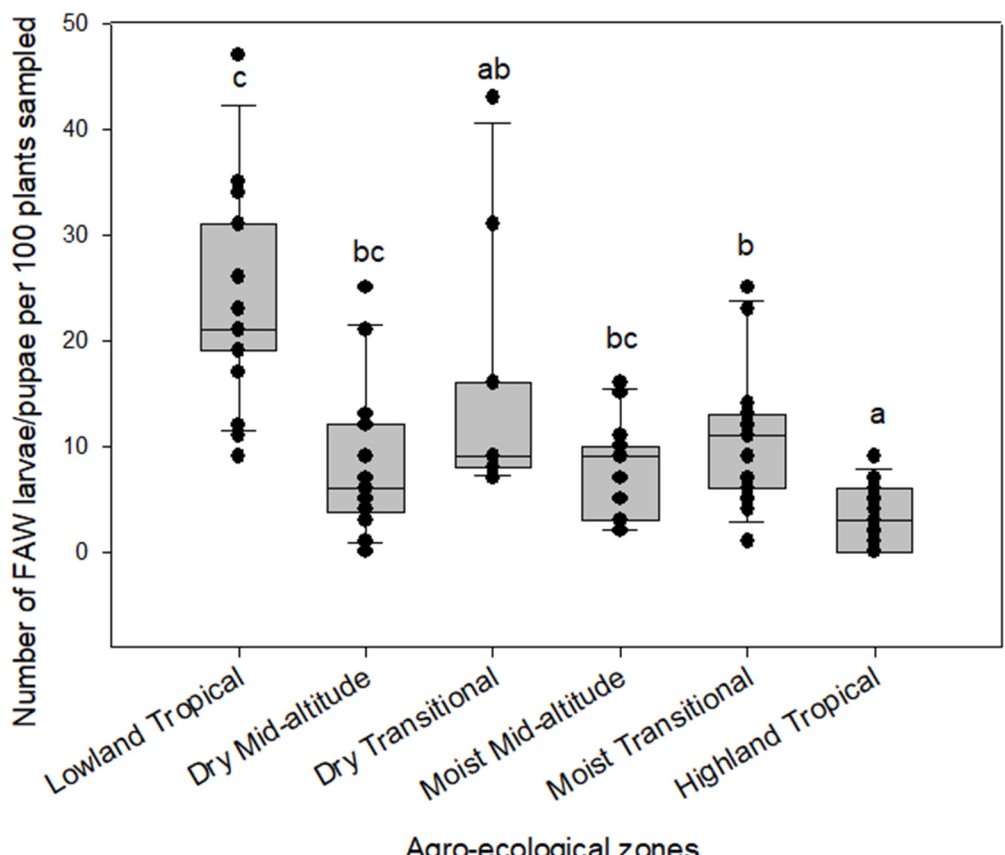

**Figure 2.** Distribution and abundance of the FAW in the different agroecological zones (AEZs) of Kenya. Non-significant differences between AEZs are shown by identical letters determined using Tukey's multiple comparisons tests with the R package "lsmeans", following the generalized linear mixed model (GLMM). Non-significant differences between AEZs are shown by identical letters.

### 3.2. Effect of FAW Introduction and Its Abundance on Stemborer Density and Parasitism across the Maize AEZs of Kenya

A total of 3543 and 2665 larvae and pupae of three stemborers, the noctuids *B. fusca* and *S. calamistis* and crambid *C. partellus*, were collected before and after the presence of the FAW, respectively, with a global total of 6208 larvae and pupae of stemborers recovered (Table A2 of Appendix A). Across the AEZs and before the presence of FAW, the maize stemborer species *C. partellus* and *S. calamistis* were recorded in lowland tropical areas, with a dominance of *C. partellus* (63.31%), while in highland tropical regions, *B. fusca* and *S. calamistis* co-occurred with a dominance of *B. fusca* (66.30%), whereas the three species were recorded in every other AEZ. The analysis of the effect of FAW introduction across different AEZs showed that the interaction between the FAW introduction and the AEZs significantly affected the stemborer density and abundance (GLMM results: $z = 1.998$, $p = 0.045676$, Supplementary Table S2). The interaction AEZs and FAW densities as a covariate also significantly reduced the abundance of stemborers (GLMM results: $z = 2.966$,

$p = 0.00301$, Supplementary Table S3). The general trend is that the density of stemborers decreased significantly in some AEZs after the arrival of FAW (Figure 3).

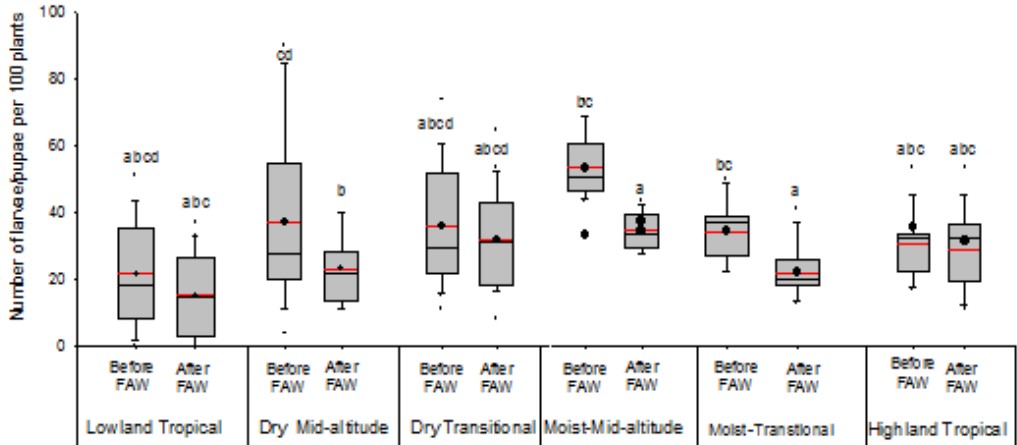

**Figure 3.** Abundance of stemborers (number of stemborer larvae and pupae per 100 plants sampled) between the period before the FAW and in the presence of the FAW across different AEZs in Kenya. Non-significant differences between AEZs are shown by identical letters determined using Tukey's multiple comparisons tests with the R package "lsmeans", following the generalized linear mixed model (GLMM). Inside each boxplot, the black line represents the median and the red line the mean. Non-significant differences between AEZs are shown by identical letters.

Furthermore, the impact of FAW after its invasion across most of the different AEZs also significantly modified the stemborer abundance at field and plant levels. At field level, both single- and multi-species infestations of maize plants were found, but their proportions of the stemborers were modified before and after FAW introduction according to the locality (Figure 4), although the maize plants sampled exhibited a similar phenological stage among AEZs during the surveys. In lowland tropical and moist mid-altitude areas, the proportions of multi-species infestation significantly increased with the presence of FAW as an additional pest in the system either at field (Figure 4A) or at plant (Figure 4B) levels ($\chi^2 = 15.07$, *df* = 1, $p = 0.0001$; $\chi^2 = 11.65$, *df* = 1, $p = 0.0006$ at field level and $\chi^2 = 7.38$, *df* = 1, $p = 0.006$; $\chi^2 = 6.39$, *df* = 1, $p = 0.01$ at plant level, respectively), while the proportions of single-species infestation significantly decreased either at field (Figure 4A) or at plant (Figure 4B) levels ($\chi^2 = 4.95$, *df* = 1, $p = 0.02$; $\chi^2 = 4.76$, *df* = 1, $p = 0.02$ at field level and $\chi^2 = 4.72$, *df* = 1, $p = 0.02$,; $\chi^2 = 3.36$, *df* = 1, $p = 0.04$ at plant level, respectively). In dry mid-altitude areas, multi-species infestation significantly increased either at field level (Figure 4A; $\chi^2 = 4.10$, *df* = 1, $p = 0.04$) or at plant level (Figure 4B; $\chi^2 = 7.73$, *df* = 1, $p = 0.005$), while in moist transitional areas, the increase of multi-species infestation was only significant at field level (Figure 4A; $\chi^2 = 10.53$, *df* = 1, $p = 0.001$). In highland tropical and dry transitional zones, no significant difference was noted ($p > 0.05$).

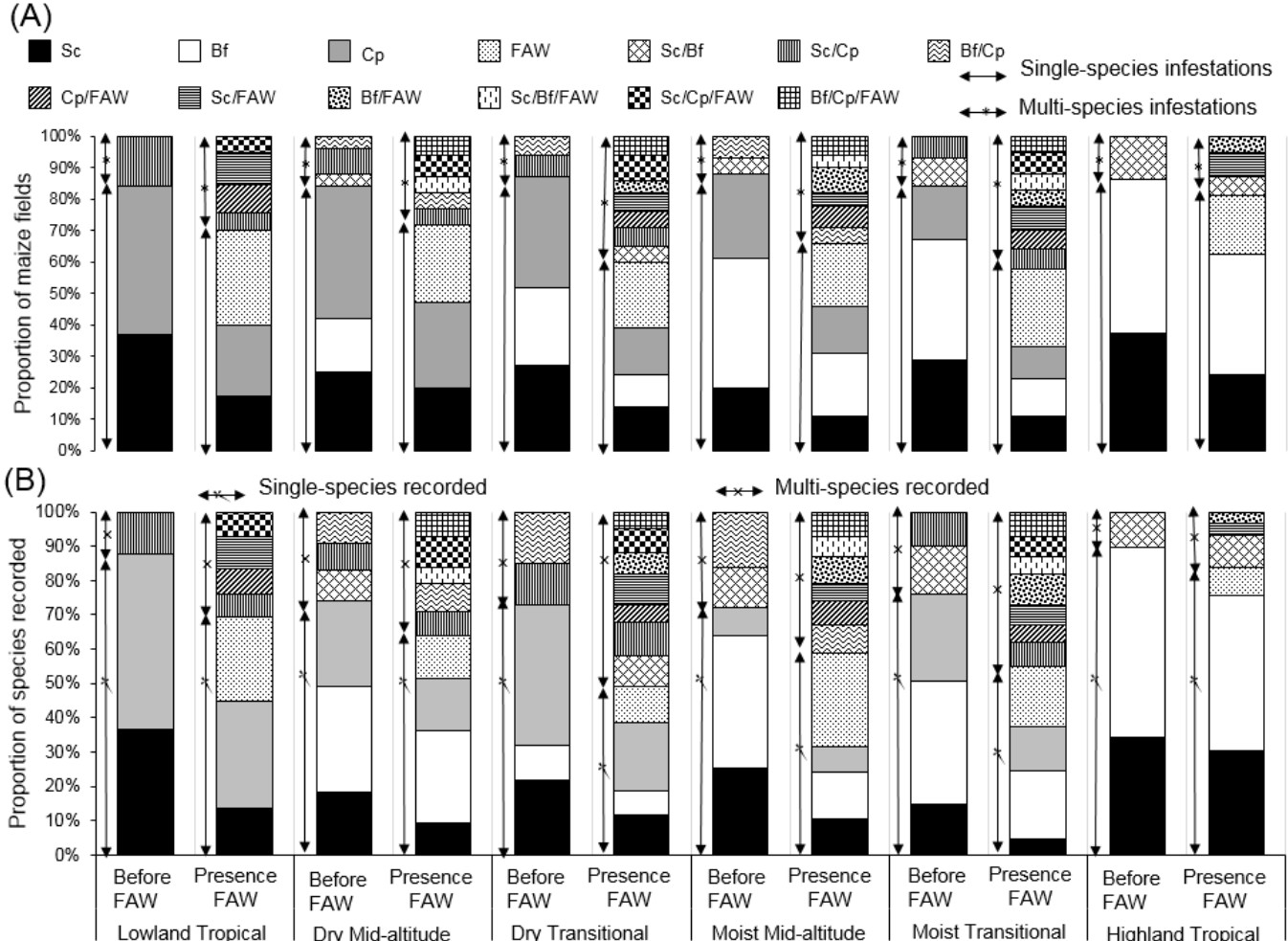

**Figure 4.** Proportion of single- and multi-species infestations at the field level (**A**) and at the plant level (**B**) of each species following the introduction of fall armyworm (FAW) in the different AEZs in Kenya. Sc = *Sesamia calamistis*; Bf = *Busseola fusca*; Cp = *Chilo partellus*; FAW = fall armyworm; Sc/Bf = *Sesamia calamistis+ Busseola fusca*; Sc/Cp = *Sesamia calamistis + Chilo partellus*; Bf/Cp = *Busseola fusca+ Chilo partellus*; Cp/FAW = *Chilo partellus+* fall armyworm; Sc/FAW = *Sesamia calamistis+* fall armyworm; Bf/FAW = *Busseola fusca+* fall armyworm; Sc/Bf/FAW = *Sesamia calamistis+ Busseola fusca* + fall armyworm; Sc/Cp/FAW = *Sesamia calamistis + Chilo partellus* + fall armyworm; Bf/Cp/FAW = *Busseola fusca+ Chilo partellus* + fall armyworm.

The parasitoid species recovered from stemborers and from FAW during the survey periods are shown in the Table 1. FAW was found to be parasitized by one species of Braconidae, two species of Tachinidae, and one species of Ichneumonidae. Parasitism was more found on stemborers than on FAW. Among all recovered parasitoids, the family of Braconidae, with *Cotesia flavipes* and *C. sesamiae*, were mostly represented and recovered from stemborers.

The overall parasitism rate of stemborers was 11.73% and 6.98% before and after the introduction of the FAW, respectively. Although stemborer parasitism significantly varied with AEZs, the impact of FAW invasion in stemborer communities significantly contributed to the decrease in stemborer parasitism observed in some AEZs after the introduction of the FAW (Figure 5) (GLMM results: $z = -2.239$, $p = 0.025$, Supplementary Table S4). The analysis of the effect of the FAW abundance coupled with AEZs, as well as their interaction, revealed a significant effect on the number of parasitized stemborer larvae (GLMM results: $z = 2.715$, $p = 0.006635$, Supplementary Table S5). The parasitism rate of each single parasitoids species from a given host species across AEZs also varied between periods before and after FAW (Supplementary Table S6).

**Table 1.** Parasitoid species recorded on stemborer species and FAW across the different AEZs before and after the presence of FAW.

| Maize Pests | Parasitoid Species | Agro-Ecological Zones | | | | | |
|---|---|---|---|---|---|---|---|
| | | Lowland Tropical | Dry Mid-Altitude | Dry Transitional | Moist Transitional | Moist Mid-Altitude | Highland Tropical |
| *Chilo partellus* | Hymenoptera: Braconidae | | | | | | |
| | *Cotesia flavipes* | x | x | x | x | x | |
| | *Cotesia sesamiae* | x | | | | | |
| | *Chelonus curvimaculatus* | | | x | | | |
| | Hymenoptera: Ichneumonidae | | | | | | |
| | *Xanthopimpla stemmator* | x | | | | | |
| | *Pediobius furvus* | | | x | | | |
| | Hymenoptera: Ceraphronidae | | | | | | |
| | *Aphanogmus fijiensis* | *x* | | | | | |
| *Sesamia calamistis* | Hymenoptera: Braconidae | | | | | | |
| | *Cotesia flavipes* | x | | x | | | |
| | *Cotesia sesamiae* | x | x | x | x | x | x |
| | *Habrobracon sp.* | | x | | | | |
| | *Dolichoginedea polaszeki* | | x | | | | x |
| | Diptera: Tachinidae | | | | | | |
| | *Siphona murina* | | x | | x | | |
| | *Descampsina sesamiae* | | x | | | | |
| *Busseola fusca* | Hymenoptera: Braconidae | | | | | | |
| | *Cotesia sesamiae* | | | x | | x | x |
| | *Dolichoginedea polaszeki* | | x | | | | |
| | Diptera: Tachinidae | | | | | | |
| | *Siphona murina* | | x | | | | x |
| | *Sturmiopsis parasitica* | | | | x | x | x |
| | Hymenoptera: Ichneumonidae | | | | | | |
| | *Xanthopimpla stemmator* | | | | | | x |
| *Spodoptera frugiperda* | Hymenoptera: Braconidae | | | | | | |
| | *Habrobracon sp.* | x | | | x | | |
| | Diptera: Tachinidae | | | | | | |
| | *Sturmiopsis parasitica* | | | | x | | |
| | *Palexorista zonata* | | x | | | | |
| | Hymenoptera: Ichneumonidae | | | | | | |
| | *Charops ater* | | | | x | x | |

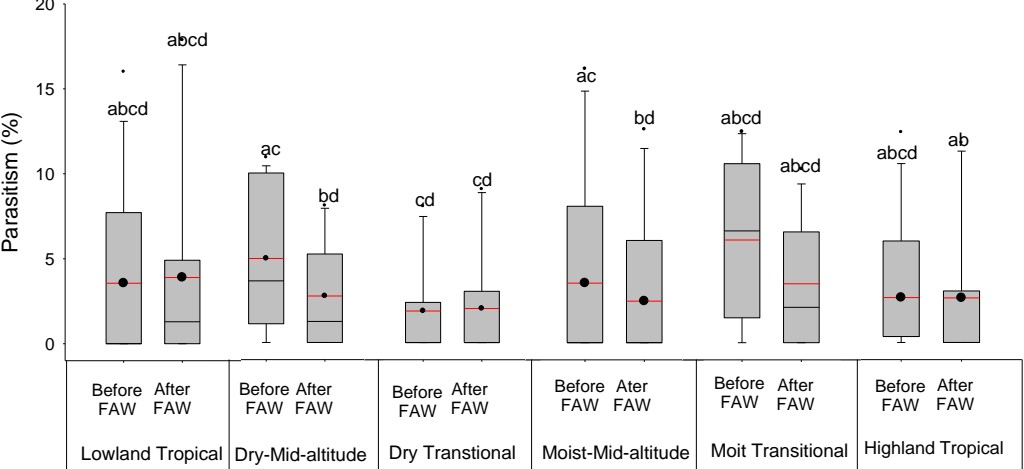

**Figure 5.** Stemborer parasitism (%) between the period before the FAW and in the presence of the FAW across different AEZs in Kenya. Non-significant differences between AEZs are shown by identical letters determined using Tukey's multiple comparisons tests with the R package "lsmeans", following the generalized linear mixed model (GLMM). Inside each boxplot, the black line represents the median and the red line the mean. Non-significant differences between AEZs are shown by identical letters.

### 3.3. Effect of FAW Introduction and Its Abundance on Stemborer Density across Different Maize Phenological Stages

The analysis of the effect of FAW introduction across crop phenological stages showed that the interaction between the FAW introduction and the host plant phenological stages significantly affected the stemborer density and abundance (GLMM results: $z = 11.77$, $p < 0.0001$, Supplementary Table S7). The interaction crop phenological stages and FAW densities as a covariate also significantly reduced the abundance of stemborers (GLMM results: $z = -3.577$, $p = 0.000348$, Supplementary Table S8).

During the pre-tasseling stage, there was no significant difference between the total number of stemborer larvae before and after the introduction of the FAW, while at both reproductive and senescence stages in maize, the total number of stemborer larvae decreased significantly when the FAW was present (Figure 6).

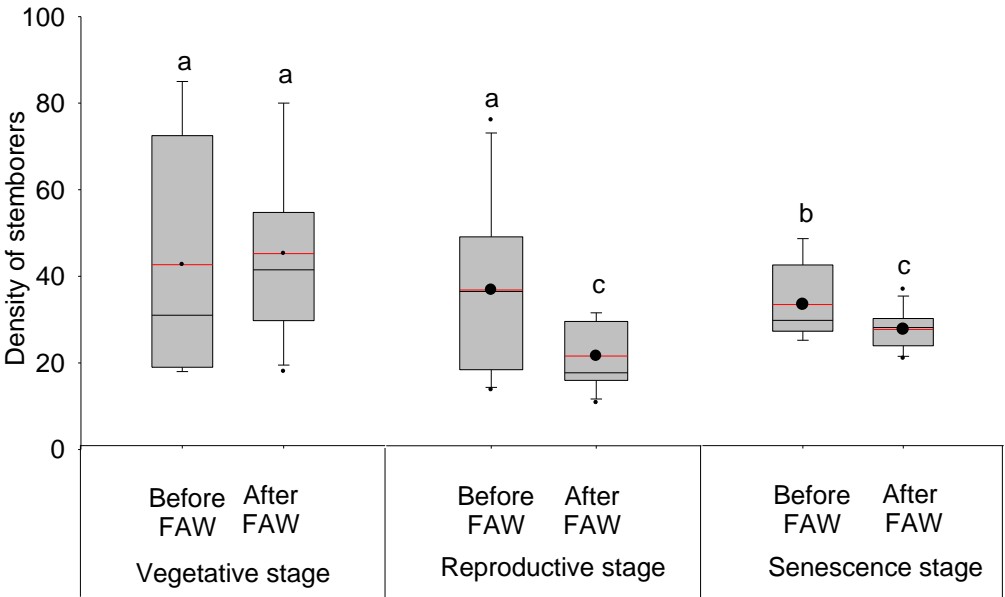

**Figure 6.** Abundance of stemborers (number of stemborer larvae and pupae per 100 plants sampled) between the period before and after the introduction of the fall armyworm (FAW) in relation to different developmental stages of maize plants in Makutano and Murang'a. Non-significant differences between AEZs are shown by identical letters determined using Tukey's multiple comparisons tests with the R package "lsmeans", following the generalized linear mixed model (GLMM). Inside each boxplot, the black line represents the median and the red line the mean. Non-significant differences between AEZs are shown by identical letters.

## 4. Discussion

The invasive maize pest, the FAW, was recorded in the six AEZs but at varying proportions between the different AEZs in this study, indicating the FAW's capacity to occupy a wide range of altitudes and environmental conditions. This is not the case for all resident stemborers; *B. fusca* is known to generally dominate the highlands and *C. partellus* the lowlands [6,7]. The capacity of an invasive species to occupy a wide range of environments, and thus, to get chance to interact significantly with native species, has been well documented [44–46].

The impact of timing of FAW (before vs. after FAW) on the densities and abundance of stem borers was studied and showed that, except in dry transitional and highland tropical areas where FAW abundance was very low, there is a correlation between FAW presence and the reduction of stemborer densities in maize fields in every other AEZ. However, FAW was not the sole factor contributing to the reduction of borer—climate factors such as temperature and rainfall, etc. were indirectly incorporated in agroecological zones and were, thus, considered in the present analysis. The presence of FAW not only



increased the multi-species infestations at field level, but also increased the proportion of multi-species larvae recorded at plant level. Therefore, the correlation between FAW presence and the reduction of stemborer densities might result in some competition between FAW and stemborer that share the same maize resource, as previously demonstrated among lepidopteran stemborers in laboratory, greenhouse [11,12], and field experiments [8], suggesting a possible displacement of stemborers by FAW. Recent field observations from Uganda highlighted the potential for displacement of stemborer populations from maize to other cereals such as sorghum by FAW [47]; moreover, the authors in [48] have reported the dominance of FAW and *C. partellus* over *B. fusca* and *S. calamistis* in multi-species systems.

In addition, the timing of FAW (before **vs**. after FAW) also negatively influenced the parasitism rate of stem borers, particularly at dry mid-altitude and moist transitional zones, although none of these parasitoids was recorded on the FAW. This reduction of parasitism was expected for two main reasons: (i) a decrease of the abundance of stemborers can render the host searching by the parasitoids' stemborer more difficult, leading to a reduction of parasitism; (ii) the cyclical relationship between populations of the pest and the associated parasitoids [48] (the parasitoids' abundance following the pests' abundance) is linked to the well-known Lotka–Volterra prey predator system, which mathematically expresses the positive correlation between pest abundance and natural enemies [49] and can explain this reduction of parasitism when the pest abundance declines. It was reported that *C. flavipes* and *C. sesamiae* species oviposit into FAW larvae under laboratory conditions, but none yielded offspring, even though they induced a significantly high nonreproductive host mortality when compared to natural mortality [50]. Furthermore, plants infested by FAW larvae are attractive to *C. flavipes* and *C. sesamiae*, and these parasitoids equally accepted the FAW and their respective native hosts [50]. Stemborer-associated parasitoids might be parasitizing unsuitable host FAW (i.e., will be unable to develop in them), investing time and energy and negatively affecting the fitness of those parasitoids [51,52]. These interferences can, therefore, have detrimental consequences on a pre-existing biological control process [51,52], explaining the overall stemborer parasitism decrease. The FAW could, therefore, represent an evolutionary trap for stemborer-associated parasitoids that undergo a reduction in their populations. Indirectly, this might later prove to be an advantage for stemborer species which might exhibit significant outbreaks due to lower demographic pressure from natural enemies [53]. In addition, since some of these parasitoids are able to parasitize FAW larvae, they might also evolve to accept the FAW larva better over time and eventually develop inside it. Other native *Cotesia* species, such as *Cotesia icipe* Fernandez-Triana & Fiaboe (Hymenoptera: Braconidae), have been found to be the dominant parasitoid of FAW larvae from field surveys done in Ethiopia, Kenya, and Tanzania [26].

Concerning host plant phenology, [54] showed that the spatial distribution of FAW larvae is random, and natural infestation is strongly associated with the maize phenological stages. Depending on the plant phenological stage, the level of infestation in maize fields increased following the introduction of the FAW. Moreover, there was no significant effect on stemborer larval abundance during maize plant pre-tasseling. However, from the maize reproductive to senescence stages, the larval abundance of stemborers decreased in the fields following the introduction of the FAW. The non-significant effect of FAW interaction with stemborers according to the maize plant phenological stages might be due to the small scale of data about plant phenology in the present study. The pre-tasseling stage of the maize plant seemed to be the most suitable host stage for the FAW. This might allow the FAW larvae, as foliar feeders, to avoid interspecific competition with stemborers, explaining the non-significant decrease of stemborer abundance at that phenological stage. Ref. [55] reported that the FAW had an infestation peak during the whorl stage of maize. After the tasseling stage, the feeding site of FAW larvae, which is essentially the central whorl, is reduced [27]. Therefore, FAW larvae can be found everywhere on the plant (tassel, silk, cob, borer's holes, etc.) [27,29], increasing the likelihood of contact and interaction between the FAW and stemborer larvae.

## 5. Conclusions

This study indicates that the timing of FAW introduction influenced the population dynamic of resident communities of maize stemborers and parasitoids. However, FAW proved to be able to co-inhabit with resident stemborers as an additional pest in maize fields across the different AEZs and different maize phenological stages. This study also suggests a possible displacement of stemborers by FAW elsewhere, for example, to other cereals. However, since this study was conducted only three years after the introduction of the FAW, further research will need to be conducted to confirm such displacements.

**Supplementary Materials:** The following are available online at https://www.mdpi.com/article/10.3390/agronomy11061074/s1, Table S1: GLMM results of distribution and abundance of the FAW in the different AEZs, Table S2: GLMM results of impact of FAW arrival on stemborer abundance in different agroecological zones, Table S3: GLMM results of effect of FAW abundance on stemborers abundance in different agroecological zones, Table S4: GLMM results of impact of FAW arrival on stemborer parasitism rates in different agroecological zones, Table S5: GLMM results of effect of FAW abundance on stemborer parasitism in different agroecological zones, Table S6: Parasitoid species recorded, and their parasitism rates found from stemborers and FAW across different AEZs, Table S7: GLMM results of impact of FAW arrival on stemborer abundance at plant phenology stages, Table S8: GLMM results of effect of FAW abundance on stemborer abundance at plant phenology stages.

**Author Contributions:** B.M.S. and P.-A.C. designed the studies. B.M.S., B.M., and J.O. collected the data. B.M.S., F.R. and E.M.A.-R. analyzed the data. B.M.S. and P.-A.C. wrote the first draft manuscript. B.M.S., P.-A.C., F.R., E.M.A.-R., S.S., D.C.K. and G.J. reviewed and approved the manuscript before submission. All authors have read and agreed to the published version of the manuscript.

**Funding:** The authors wish to thank the German Academic Exchange Service (DAAD) for funding the PhD fellowship under the grant number 91636630 from the University of Nairobi and the *icipe* Capacity Building Program (ARPPIS) for hosting the PhD student. This research was funded by the "Institut de Recherche pour le Développement" (IRD), France through the Noctuid Stemborers Biodiversity (NSBB) project, grant number (B4405B) and the Integrated pest management strategy to counter the threat of invasive fall armyworm to food security in eastern Africa (FAW-IPM) (grant number: DCI-FOOD/2017/) financed by the European Union. We also acknowledge the financial support for this research by the following organizations and agencies: the UK's Department for International Development (DFID); the Swedish International Development Cooperation Agency (Sida); the Swiss Agency for Development and Cooperation (SDC); and the Kenyan Government.

**Acknowledgments:** Our thanks to Fritz Schulthess for his critical review of the manuscript and to Malcolm Eden for his English corrections.

**Conflicts of Interest:** The authors declare no conflict of interest.

## Appendix A. Distribution and Abundance of the FAW and Stemborers in the Different AEZs of Kenya

**Table A1.** Number (mean ± SE) of larvae and pupae of *Busseola fusca*, *Sesamia calamistis*, *Chilo partellus*, and *Spodoptera frugiperda* per 100 maize plants sampled in each maize field before and after the presence of FAW in different AEZs.

| AEZs | Species | Before FAW | After FAW | Likelihood Ratio (LR) | $z$-Value | $p$-Value |
|---|---|---|---|---|---|---|
| Lowland tropical | *Chilo partellus* | 17.54 ± 2.81 a | 13.67 ± 2.52 a | 0.77 | −0.87 | 0.38 |
| | *Sesamia calamistis* | 4.40 ± 0.86 b | 1.6 ± 0.28 a | 12.84 | 3.48 | 0.0003 |
| | *Spodoptera frugiperda* | 0.00 ± 0.00 a | 19.54 ± 2.44 b | 217.59 | −0.006 | <0.0001 |
| Dry mid-altitude | *Busseola fusca* | 4.53 ± 0.75 a | 2.70 ± 0.56 a | 3.69 | 1.91 | 0.05 |
| | *Sesamia calamistis* | 10.00 ± 1.74 a | 6.53 ± 0.92 a | 3.06 | 1.74 | 0.07 |
| | *Chilo partellus* | 22.80 ± 5.11 a | 14.11 ± 1.80 a | 3.70 | 1.92 | 0.05 |
| | *Spodoptera frugiperda* | 0.00 ± 0.00 a | 6.76 ± 1.44 b | 71.72 | −0.004 | <0.0001 |
| Dry transitional | *Busseola fusca* | 28.94 ± 3.18 a | 23.41 ± 3.98 a | 1.32 | 1.15 | 0.24 |
| | *Sesamia calamistis* | 4.80 ± 1.54 a | 3.73 ± 0.53 a | 0.3713 | 0.610 | 0.54 |
| | *Chilo partellus* | 1.67 ± 0.49 a | 2.20 ± 0.45 a | 0.553 | −743 | 0.45 |
| | *Spodoptera frugiperda* | 0.00 ± 0.00 a | 12.41 ± 2.34 b | 85.31 | −7.84 | <0.0001 |

**Table A1.** *Cont.*

| AEZs | Species | Before FAW | After FAW | Likelihood Ratio (LR) | z-Value | p-Value |
|---|---|---|---|---|---|---|
| Moist mid-altitude | *Busseola fusca* | 8.40 ± 2.18 a | 7.20 ± 0.86 a | 0.2301 | 0.480 | 0.63 |
| | *Sesamia calamistis* | 17.15 ± 1.68 b | 11.52 ± 1.59 a | 5.94 | 2.44 | 0.01 |
| | *Chilo partellus* | 26.73 ± 2.84 b | 16.05 ± 1.90 a | 11.89 | 3.46 | 0.0005 |
| | *Spodoptera frugiperda* | 0.00 ± 0.00 a | 7.66 ± 1.15 b | 107.43 | 0.003 | <0.0001 |
| Moist transitional | *Busseola fusca* | 9.80 ± 1.25 a | 9.86 ± 1.33 a | 0.0015 | −0.039 | 0.96 |
| | *Sesamia calamistis* | 7.92 ± 1.56 a | 5.54 ± 1.42 a | 1.1178 | 1.065 | 0.29 |
| | *Chilo partellus* | 7.13 ± 1.12 a | 6.60 ± 1.09 a | 0.12 | 0.35 | 0.72 |
| | *Spodoptera frugiperda* | 0.00 ± 0.00 a | 10.80 ± 1.69 b | 122.11 | −0.003 | <0.0001 |
| Highland tropical | *Busseola fusca* | 20.93 ± 2.76 a | 19.00 ± 2.70 a | 0.071 | 0.267 | 0.78 |
| | *Sesamia calamistis* | 10.13 ± 1.24 a | 9.33 ± 0.82 a | 0.287 | 0.536 | 0.59 |
| | *Spodoptera frugiperda* | 0.00 ± 0.00 a | 3.33 ± 1.00 b | 26.698 | −0.004 | <0.0001 |

Line comparisons: non-significant differences of the density of each species between the two periods (prior FAW and presence FAW) are shown by identical letters determined using Tukey's multiple comparisons tests with the R package "lsmeans," following the generalized linear model (GLM) with negative binomial error distribution.

**Table A2.** Total number (relative proportions (%) are given in parenthesis) of larvae and pupae of *Busseola fusca*, *Sesamia calamistis*, *Chilo partellus*, and *Spodoptera frugiperda* per 100 maize plants sampled in each maize field before and after the presence of FAW in different AEZs.

| AEZs | Species | Before FAW | After FAW | Total Number |
|---|---|---|---|---|
| Lowland tropical | *Chilo partellus* | 383 (63.31) | 386 (41.15) | 769 (49.84) |
| | *Sesamia calamistis* | 222 (36.69) | 122 (13.01) | 344 (22.29) |
| | *Spodoptera frugiperda* | - | 430 (45.84) | 430 (27.87) |
| Dry mid-altitude | *Busseola fusca* | 68 (12.14) | 46 (8.98) | 114 (10.63) |
| | *Sesamia calamistis* | 150 (26.79) | 111 (21.68) | 261 (24.35) |
| | *Chilo partellus* | 342 (61.07) | 240 (46.88) | 582 (54.29) |
| | *Spodoptera frugiperda* | - | 115 (22.46) | 115 (10.73) |
| Dry transitional | *Busseola fusca* | 550 (39.74) | 398 (36.92) | 948 (38.51) |
| | *Sesamia calamistis* | 326 (23.55) | 196 (18.18) | 522 (21.20) |
| | *Chilo partellus* | 508 (36.71) | 273 (25.32) | 781 (31.72) |
| | *Spodoptera frugiperda* | - | 211 (19.57) | 211 (8.57) |
| Moist mid-altitude | *Busseola fusca* | 126 (56.50) | 108 (34.29) | 234 (43.49) |
| | *Sesamia calamistis* | 72 (32.44) | 56 (17.78) | 128 (23.79) |
| | *Chilo partellus* | 25 (11.21) | 33 (10.48) | 58 (10.78) |
| | *Spodoptera frugiperda* | - | 118 (37.46) | 118 (21.93) |
| Moist transitional | *Busseola fusca* | 147 (45.94) | 148 (34.18) | 295 (39.18) |
| | *Sesamia calamistis* | 66 (20.63) | 24 (5.54) | 90 (11.95) |
| | *Chilo partellus* | 107 (36.71) | 99 (22.86) | 206 (27.36) |
| | *Spodoptera frugiperda* | NA | 162 (37.41) | 162 (21.51) |
| Highland tropical | *Busseola fusca* | 299 (66.30) | 285 (60.00) | 584 (63.07) |
| | *Sesamia calamistis* | 152 (33.70) | 140 (29.47) | 292 (31.53) |
| | *Spodoptera frugiperda* | NA | 50 (10.53) | 50 (5.40) |
| | Total number | 3543 | 3751 | 7294 |

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
