# Peer review of "Impact of an Exotic Invasive Pest, Spodoptera frugiperda (Lepidoptera: Noctuidae), on Resident Communities of Pest and Natural Enemies in Maize Fields in Kenya"

_agronomy, doi:10.3390/agronomy11061074_

Round 1

Reviewer 1 Report

This manuscript showcases the impacts of an invasive species on locally-occurring insects in maize. This study will add information on the vast literature on biological invasions. Also, this manuscript provides critical information for the understanding of how the introduction of FAW is affecting the current borer community in Kenya maize.

I am including a PDF file detailing all my edits, corrections and comments, using the Comment function.

I have two major concerns on the current version of this manuscript, regarding: 1) data analysis and 2) Discussion section. Authors are consistently reporting single effects, even though interactions are significant. It is also unclear how authors analyzed FAW presence and stem borers. It seems authors used FAW as a fixed factor; however, there is a time component (before- and after- infestation) that authors will need to provide more details on the analysis.

Authors also missed the opportunity to include in the discussion the impact of timing of FAW on the densities of stem borers. I am proposing to have this missing information as a section at the beginning of the Discussion section.

Author Response

Reviewer#1

This manuscript showcases the impacts of an invasive species on locally-occurring insects in maize. This study will add information on the vast literature on biological invasions. Also, this manuscript provides critical information for the understanding of how the introduction of FAW is affecting the current borer community in Kenya maize. 

I am including a PDF file detailing all my edits, corrections and comments, using the Comment function.

I have two major concerns on the current version of this manuscript, regarding: 1) data analysis and 2) Discussion section. Authors are consistently reporting single effects, even though interactions are significant. It is also unclear how authors analyzed FAW presence and stem borers. It seems authors used FAW as a fixed factor; however, there is a time component (before- and after- infestation) that authors will need to provide more details on the analysis.

Authors: In the GLMM analyses done and detailed in Mat. & Met. section, among the fixed factors we have not only FAW (which is after its arrival only since before there is no FAW) but also FAW (before/after) including somehow the time component (before and after). This is mentioned in the text.

The following GLMM models were performed:
1. FBW abundance (from years after introduction) ~ AEZ + (locality/field)
2. Stemborer abundance ~ Before/After FAW + AEZ + Before/After FAW:AEZ+(locality/field)
3. Stemborer abundance ~ FAW abundance (from years after introduction) +AEZ +FAW abundance :AEZ+(locality/field)
4. Stemborer parasitism ~ Before/After FAW + AEZ +Before/After FAW:AEZ+(locality/field)
5. Stemborer parasitism ~ FAW abundance (from years after introduction) +AEZ +FAW abundance :AEZ+(locality/field)
6. Stemborer Density ~ Before/After FAW + plant stage +Before/After FAW:plant stage+(locality/field)
7. Stemborer Density ~  FAW abundance (from years after introduction) + plant stage +FAW abundance :plant stage+(locality/field)

Authors also missed the opportunity to include in the discussion the impact of timing of FAW on the densities of stem borers. I am proposing to have this missing information as a section at the beginning of the Discussion section.

Authors: Actually, the overall discussion was around the impact of the timing of FAW on the densities of stem borers (before vs. after FAW)? We tried to make more clearly in the text that this was the focus point of the discussion.

Kindly find our point-by-point replies:

Rev.: I do not follow how authors analyzed this data set. I would have assumed to use some sort of a repeated measure analysis, where stem borers densities before FAW were compared against densities after FAW. Did the authors used FAW densities as a covariate, making it a fixed factor? Also, it is expected that there will be a regional difference among taxa; so why incorporating region into the analysis?  My suggestion is to analyze data by region separately and compare the trends.

Authors: The data analyses are detailed in Mat. & Met: GLMM with a Poisson distribution was performed using two levels of analysis: (i) the impact of FAW arrival on stemborer abundance and their parasitism, where the FAW (before/after) and AEZs were the fixed effects and Fields/Locations were random effects of field nested within locality; (ii) how the FAW abundance affects stemborers and their parasitism, where FAW abundance (after FAW invasion only) and AEZs were the fixed effects and Fields/Locations were random effects.

All analyses have been done by AEZs separately somehow already by region separately.

Rev.: Single effects should not have been reported since interaction was significant. Interaction needs to be sliced, to control for single effects.

Again, the interaction is significant. Single effects can not be interpreted separately.

Authors: On the text, now only the results of interactions of the GLMM are given and all tables are removed as suggested by the reviewer in Appendix. Instead we have inserted the raw data at Table A1 and A2

Rev. This is the correct figure to analyze the interaction. On the X-axis, before and after FAW data sets are presented. So, there is a time component in the analysis? If yes, how did the authors control for field localities, potentially being different within each AEZ?

Authors: Responded in Mat & Met section by: “Sampling in same selected locations was done between 2012 and 2016, before the introduction of the FAW, and between 2018 and 2019, after the introduction of the FAW.”

Rev. I am surprised that authors have not discussed, at all, the impact of the timing of FAW on the densities of stem borers (before vs. after FAW).

Authors: Actually, the overall discussion was around the impact of the timing of FAW on the densities of stem borers (before vs. after FAW)? We tried to make more clearly in the text that this was the focus point of the discussion.

Rev. Could authors calculate the ratio between stem borers and FAW? Does FAW dominate the infestation on a maize plant? Did author find more FAW larvae from each inspected plant, when compared to the stem borers?

Authors: A new figure (cf figure 4 in the revised version) with a text in the results section are responding to this remarck

Reviewer 2 Report

Minor comments:

The number of sampling in this investigation is quite high, which is appreciated and results in accuracy of the investigation.

  • Abstract, line 25-26: Results are not well presented in the abstract. It’s just mentioned that FAW significantly affected the maize stemborer, but how affected? I suggest presenting the results more clearly here.

  • Results, Line 192-193: In the text is written that FAW abundance had a significant effect on stemborer abundance across different zone. However, there is no significant difference in Dry transitional and highland tropical based on figure3. I suggest editing the text accordingly.

  • Results, Line 200-213: As density of stemborers decreased after introduction of the FAW, therefore, it is not surprising that parasitism decreased as well. Is this percentage of parasitism presented here because of the direct effect of FAW presence or it is because of lower density of stemborers? How do you explain this? I suggest this issue to be interpreted at the discussion.

  • Discussion: Did you find any parasitoids on FAW?

  • Line 272-273: This study was conducted in the lab or it’s the result of filed work? Please mention it in the text.

Good Job!

Author Response

Reviewer#2

Minor comments:

The number of sampling in this investigation is quite high, which is appreciated and results in accuracy of the investigation.

  • Abstract, line 25-26: Results are not well presented in the abstract. It’s just mentioned that FAW significantly affected the maize stemborer, but how affected? I suggest presenting the results more clearly here.

 Done

  • Results, Line 192-193: In the text is written that FAW abundance had a significant effect on stemborer abundance across different zone. However, there is no significant difference in Dry transitional and highland tropical based on figure3. I suggest editing the text accordingly.

 Done

  • Results, Line 200-213: As density of stemborers decreased after introduction of the FAW, therefore, it is not surprising that parasitism decreased as well. Is this percentage of parasitism presented here because of the direct effect of FAW presence or it is because of lower density of stemborers? How do you explain this? I suggest this issue to be interpreted at the discussion.

 Now this aspect is discussed.

  • Discussion: Did you find any parasitoids on FAW?

Results are now given on what parasitoids have been found on FAW.

  • Line 272-273: This study was conducted in the lab or it’s the result of filed work? Please mention it in the text.

Done

Round 2

Reviewer 1 Report

I still believe that this manuscript is in need of further revisions. Authors did a phenomenal job addressing proposed edits and comments. However, those edits did not translated into the manuscript. Authors provided firm and concise answers in their response letter, but they are still missing to incorporate that provided information into the manuscript.

I am attaching a PDF with my edits and comments. Areas that authors will need to work on are:

1) The phrasing of presenting results. I do understand there is an impact of the presence of FAW on stemborers; however it may not be the only reason why population dynamic of these borers are changing. As it is now, the manuscript is presenting FAW as the solely factor contributing to the reduction of borer. It would be ideal that authors can make readers aware that there might be other factors affecting stemborers populations (environmental conditions, maize area planted, and so forth).

2) Include additional information on box plots

3) Include a discussion point on the proportion of stemborers and FAW results generated from Fig. 4.

Author Response

Responses to the reviewer

  1. I still believe that this manuscript is in need of further revisions. Authors did a phenomenal job addressing proposed edits and comments. However, those edits did not translated into the manuscript. Authors provided firm and concise answers in their response letter, but they are still missing to incorporate that provided information into the manuscript.

  1. I am attaching a PDF with my edits and comments. Areas that authors will need to work on are:

  • The phrasing of presenting results. I do understand there is an impact of the presence of FAW on stemborers; however it may not be the only reason why population dynamic of these borers are changing. As it is now, the manuscript is presenting FAW as the solely factor contributing to the reduction of borer. It would be ideal that authors can make readers aware that there might be other factors affecting stemborers populations (environmental conditions, maize area planted, and so forth).
  • Response: FAW was not as the solely factor contributing to the reduction of borer but there is also climate factors such as temperature and rainfall, etc. together indirectly incorporated in agroecologiocal zones considered in the present analysis. This is now mentioned into the discussion.
    •  

  • Include additional information on box plots

Response: additional information added to boxplot legends

3) Include a discussion point on the proportion of stemborers and FAW results generated from Fig. 4.

  • Response: This has been inserted in the discussion now.                      

Specific comments:

  1. Lines 41-42: Estimated crop losses due to these pests vary with agro-ecological zones (AEZs). Response: The was sentence deleted.
  2. Why not including F value and degrees of freedom (DF) in GLMM results? Response: The summary output of Generalized Linear Mixed Model (GLMM) does not generate either F value or degree of freedom, mostly where fall armyworm density was used as covariate. However, we now provided the summary of each GLMM performed in supplementary material.
  3. Line 178: “(mean±SE)”: Response: now deleted
  4. Figure 4: Why are these bars truncated? It seems actors used the value '0' as the cut-off while plotting these graphs. Response: Bars were not truncated. As you can see in the current Figure 4, the value '-1' is now used as the cut-off while plotting these graphs. So, the forms of the bars is depended on the nature of the data.
  5. Table 1: To complement the information provided by this table, authors can include the actual number of the parasitoids eclosing from each of the hosts. Response: The number of the parasitoids eclosing from each of the hosts was not recovered, but the number of the parasitized hosts out of the total of a given host species. Details on the parasitism rates are now given in Supplementary Table S6.